# Factors Influencing Students' Continuance Intention to Learn in Blended Environments at University

Tatiana Baranova, Aleksandra Kobicheva * and Elena Tokareva

Institute of Humanities, Peter the Great Saint-Petersburg Polytechnic University, 195251 St. Petersburg, Russia; baranova_ta@spbstu.ru (T.B.); tokareva_eyu@spbstu.ru (E.T.)
* Correspondence: kobicheva92@gmail.com

**Abstract:** (1) Background: The main goal of this study was to determine the factors that have an influence on the continuance intention to learn in blended environments. (2) Methods: For our study, we created a research model based on the Expectation–Confirmation Model (ECM) and Theory of Planned Behavior (TPB), which involves a total of seven latent constructs and contains a total of eight hypotheses. A total of 301 undergraduate and postgraduate students studying at Humanity Institute of Peter the Great Polytechnic University voluntarily participated in the study. The online survey consisted of 22 items that determined the seven indicators studied and was conducted in the spring semester of 2021. For our analysis and hypothesis testing, we used PSS 24.0 and SmartPLS 3.0 programs. (3) Results: According to the results of this study, all the proposed hypotheses were confirmed, which confirmed the influencing power of research model indicators. Also, it was revealed that such indicators as confirmation and attitude are the key factors that affect the continuance intention to learn in a blended environment. (4) Conclusions: As the COVID-19 pandemic is an ever-changing situation, it is important to understand student perceptions of blended learning and manage their continuance intention to learn in such environments. This study contributes to such knowledge and provides insightful implications for academia.

**Keywords:** distance education; e-learning; teaching COVID-19; student continuance intention; blended learning

## 1. Introduction

The devastating nature of the COVID-19 pandemic has affected almost every sector of society in the world, and higher education is no exception [1]. The COVID-19 pandemic has disrupted most existing practices of teaching and forced the teaching and learning process to change unpredictably and quickly [2]. In the 2020/21 academic year, the vast majority (88.5%) of universities have ended up adopting a "blended learning" approach, and Russian universities are not exceptions. For instance, in Peter the Great St. Petersburg Polytechnic University, all lectures were provided in live online-learning format while seminars were in classes. According to Student Crowd [3], although institutions, such as Oxford and Cambridge in particular, have dedicated themselves entirely to online learning, each format has unique benefits and creates individual challenges for teaching and learning [4–11]. However, the extent to which these modes affect students' perceptions of higher education remains unclear.

For our study, we used the Expectation–Confirmation Model [12] and Theory of Planned Behavior [13] as a framework for studying the factors influencing university students' acceptance of blended learning and the relationships between these factors.

This study is relevant because the implementation of blended learning is considered from a variety of perspectives, including attitudes toward blended learning, subjective norms, perceived behavioral control, satisfaction, validation, and perceived utility. It means universities should look into these factors before implementing blended learning. This study contributes to such knowledge and provides insightful implications for academia.

The current study describes an investigation of university students' continuance intention to learn in blended environments in the academic year 2020–2021. Factors influencing students' continuance intention to learn in blended format were analyzed and presented. Thus, the paper is based on two research questions:

1.  What underlying factors contribute to students' continuance intention to learn in blended environment at university?
2.  Do these underlying factors significantly and positively influence students' intention to continue their learning in blended format?

The rest of this paper is organized as follows. Section 1.1 describes the theoretical background of the research; the integrated research model and research hypothesis are presented. In Section 2, the methodology is described, with the demographic profile of respondents illustrated. In Section 3, the validity testing is presented. In addition, factors influencing the intention of students to continue learning in blended environments are analyzed and discussed. Finally, Section 4 presents the conclusion and future research directions.

### 1.1. Theoretical Background

#### 1.1.1. Blended-Learning

Virtual learning is seen as learning that takes place effectively outside the traditional classroom environment [14]. Previous studies have shown the anti-role benefits and drawbacks of online learning. While undergraduate students require face-to-face contact with instructors in order to adequately assimilate the required knowledge [6,15], many students reacted well to virtual learning, positively noting the comfort of the learning environment and the possibility of effective time management through the repetition of video content [10,11].

Blended learning is a combination of virtual learning and face-to-face learning [16–18]. Research by some scholars has identified the benefits of blended learning, among which are the flexibility of higher education [11], increased student engagement, and the development of self-control and regulation of the learning process [19–22].

In Hrastinski [23], the most commonly used definitions of blended learning are given by Graham [24] (p. 5): "blended learning systems combine face-to-face learning with machine learning" and Garrison and Kanuka [25] (p. 97): "Thoughtful integration face-to-face teaching in a classroom with an online learning experience." During the COVID-19 epidemic, almost all universities introduced an online learning system that provides access to, at least, educational materials. In this regard, blended learning is also called the "new traditional model" or "new normal" [26].

Other authors consider blended learning according to the proportional relationship between virtual and face-to-face learning. Bernard, Borochowski, Schmid, Tamim, and Abrami [27], for example, used both virtual and traditional learning equally in their analysis. This view is shared by several other studies, based on the fact that in blended learning, students prefer the large areas of learning offered online [28–30]. There is also an opposing view that blended learning with additional integration of online components can lead to an increased workload for students and teachers compared to a traditional course [31].

#### 1.1.2. Intention to Learn

To explain the adoption and continued use of technology in education, researchers look at motivational indicators [32]. It should be taken into account that the technologies introduced into the educational process were not necessarily used constantly; that is, situations are acceptable when users initially accepted the technology, but later stopped using it. It is situations like these that give relevance to the study of consumer behavior and what influences the transition from the introduction of technologies to the abandonment of them. The most representative research models are the expectation–confirmation model (ECM) and the theory of planned behavior [12]. The application of these theories contributes to the study of the psychological mechanism that encourages users to continue using technology.

### 1.1.3. The Expectation–Confirmation Model (ECM)

The Expectation–Confirmation Model (ECM) is widely used as an exploratory information systems (IS) model. This model focuses on user satisfaction and their intention to continue using information systems [12]. Several papers have been reviewed that study the ECM model, as well as use various variations of such a model [33,34]. In these studies, the following variables were identified: confirmation, perceived utility, and satisfaction. These variables were considered as key factors in explaining the intentions of users who want to resume the use of electronic technologies in education. Also, these factors tested whether the initial expectations were confirmed after training using this method.

For this study, of particular interest are works that study the intention to continue learning. For example, Ouyang et al. [35] used the ECM as a base model to examine the intention of Chinese students to continue learning.

Some researchers did not dwell on the classical model of expectation confirmation but supplemented it with important components that are relevant for the purposes of their research. Thus, Zhou [34] expanded the ECM by adding the construct "social influence" and learning outcomes, thus replacing perceived usefulness in the context of learning. In their study, Alraimi, Zo, and Cyganek [36] extended the ECM using data from students enrolled in three major educational platforms—Coursera, EdX, and Udacity. In this study, an intrinsic motivation variable is added to the ECM model, taking into account "perceived pleasure", "perceived openness", and "perceived reputation".

ECM assumes that technology users make cognitive comparisons when making decisions about continued use. This process of cognitive comparison is well-described in the famous marketing study Oliver's Cognitive Model [37]. This theory describes the assumption about the expectations of users that they form for each product, based on various sources of information. Such sources of information for users may be recommendations from acquaintances, the media, or past experience with similar products or services. User expectations are formed before using a particular product. Depending on the extent to which the initial expectations are confirmed in the process of using the product, the level of user satisfaction will be formed, which in the future will be reflected in their decision to continue using or reusing the product. In the theory under consideration, satisfaction is understood as the emotional evaluation of a product by a person. Nonconfirmation, as the most important intermediate variable in the marketing literature [38], conceptualizes the perceived discrepancy between previous expectations and perceived performance.

Based on Oliver's cognitive model, ECM actualizes two important variables: satisfaction and confirmation. Bhattacherjee [12] argued that confirmation and satisfaction already take into account the construct of "perceived performance". Therefore, it was removed from the model. In addition, it is taken into account that experience has the ability to influence and change expectations. This is justified by the fact that in the cognitive memory of each individual, after the fact, the expectation will take the place of the original expectation. It can be concluded that it is the post factum expectation that will ultimately have an impact on the process of forming intentions. Finally, ex post expectations in the model were contextualized in the I/S study. The perceived usefulness of the Technology Acceptance Model (TAM) [39] in a technology use situation will highlight post hoc expectation as one of the core cognitions.

This study is consistent with ECM [12] in that the level of confirmation is critical to explain intent to continue, and furthermore, replacing initial expectations with ex post expectations (represented by perceived utility) makes the model more representative as it gets closer to the next one.

### 1.1.4. Theory of Planned Behavior (TPB)

The theory of planned behavior (TPB) proposed by Ajzen [40] suggests that behavior is directly influenced by behavioral intention, which in turn depends on three motivational factors: attitude, subjective norm, and perceived behavioral control. Behavioral intention is a suitable metric to measure actual behavior [41].

To determine such a variable as an attitude, an assessment of the results of behavior according to the expectations of a person and whether these results are desirable or not [13,40], in other words, a person's assessment of the advantages or disadvantages of certain behavior, is used [42]. Depending on how the result of behavior is assessed by a person (as satisfactory or unsatisfactory), positive or negative attitudes are created accordingly. Subsequently, these attitudes influence behavioral intention.

A subjective norm is an indicator that describes how a person perceives social pressure, how much it encourages them to behave in a certain way. Social pressure includes relationships and expectations of significant others. The subjective norm, in turn, influences behavioral intentions. Thus, depending on the degree of approval of others, a person regulates their behavior [42].

Perceived behavioral control is a measure based on how people self-assess their ability to control certain behaviors [43]. Perceived behavioral control depends on a person's belief in their ability to perform a particular behavior, as well as their resources. According to research, perceived behavioral control affects not only behavioral intentions, but also behavior itself [44].

We believe that the use of TPB is effective for studying the causes and predicting behavior, since this theory reflects in the model the main factors of personal (relation-ships and perceived behavioral control) and social (subjective norm) influence. TPB has been applied in a variety of contexts such as technology, healthcare, and politics [39,45–47].

1.1.5. Research Framework and Hypothesis

When forming the research model in this work, the following points were taken in-to account.

First, in the context of the current study, we distinguish between the following concepts—perceived behavioral control and attitude. As Aizen [44] has already pointed out, perceived behavioral control is not directly related to the likelihood of performing a particular behavior and achieving a particular outcome. We consider perceived behavioral control as a subjective measure of the degree of behavioral control. For the current study, perceived behavioral control reflects how students perceive blended learning, whether they find it easy or difficult.

Second, we use intent instead of actual behavior. Aizen [44] (p. 181) states that "intentions are assumed to reflect motivational factors influencing behavior." Thus, we hypothesize that the degree of intention to perform a behavior increases the likelihood of the actual behavior. Venkatesh and Davis [48] and Venkatesh, Morris and Ackerman [49] confirm that intention and actual behavior do not match. This is due to the abrupt transition to blended learning due to the emergence of the COVID-19 pandemic and the unpredictable development of the disease. Therefore, actual behavior may lead to incorrect conclusions. Thus, we use behavioral intent because it is the immediate precursor to actual behavior [44].

Thirdly, we distinguish between constructs—satisfaction and attitude. It is assumed that satisfaction in ECM directly affects the formation of the intention to continue using [12]. Bhattacharjee agreed with LaTour and Peat [50] that satisfaction and attitude are inherently synonymous. It is important to note the difference between them, which is the measurement time. Attitude is a construct prior to acceptance, while satisfaction is considered after acceptance. It is worth noting that some papers indicate a conceptual difference between these measures [37,51]. For example, satisfaction reflects the emotions received after use, while attitude reflects the affective response to behavior [51]. In addition, satisfaction is based on past user experiences [32]. Attitude, on the other hand, is focused on feelings about the future experience of use. Thus, we can conclude that satisfaction and attitude are theoretically different constructs.

We measured students' intention to continue learning in a blended environment using the following six key indicators: attitude, subjective norms, perceived behavioral control [52], perceived usefulness, confirmation, and satisfaction [53,54]. Figure 1 presents

the integrated research model of the factors contributing to the student's continuance intention to learn in blended environment.

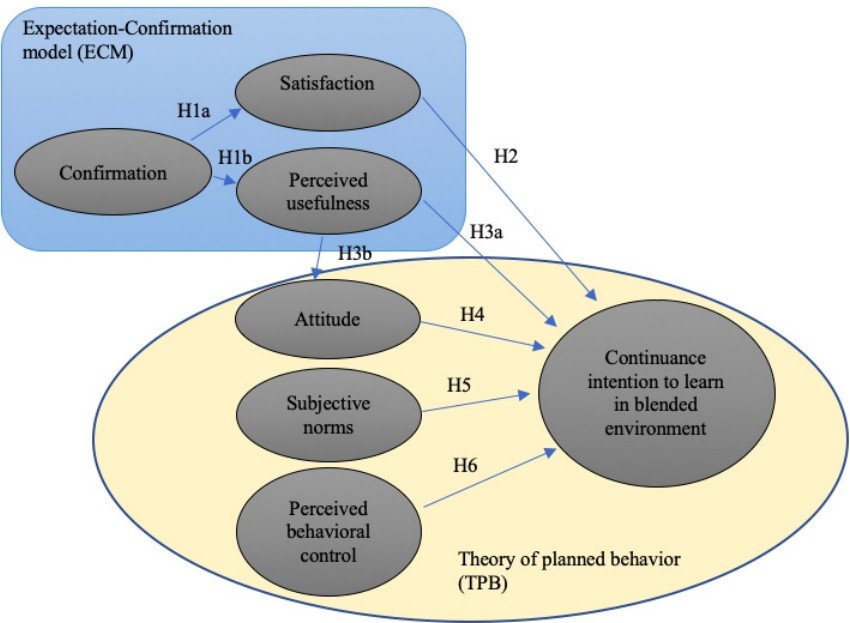

**Figure 1.** The research framework of the study.

To predict students' continuance intention toward blended learning environment through an integration of ECM and TPB, we put forward eight hypotheses:

**Hypothesis 1a (H1a).** *Confirmation significantly and positively affects satisfaction.*

**Hypothesis 1b (H1b).** *Confirmation significantly and positively affects perceived usefulness.*

**Hypothesis 2 (H2).** *Satisfaction significantly and positively affects continuance intention.*

**Hypothesis 3a (H3a).** *Perceived usefulness significantly and positively affects continuance intention.*

**Hypothesis 3b (H3b).** *Perceived usefulness significantly and positively affects attitude.*

**Hypothesis 4 (H4).** *Attitude significantly and positively affect continuance intention.*

**Hypothesis 5 (H5).** *Subjective norms significantly and positively affect continuance intention.*

**Hypothesis 6 (H6).** *Perceived behavioral control significantly and positively affects continuance intention.*

Figure 1 displays our research model, which involves a total of seven latent constructs and consists of eight hypotheses. The initial constructs and links of expectation–confirmation model (ECM) are indicated in the blue box; the original variables and paths of theory of planned behavior (TPB) are highlighted in the yellow circle.

## 2. Materials and Methods

A total of 301 undergraduate and postgraduate students studying at Humanity Institute of Peter the Great Polytechnic University voluntarily participated in the study. Table 1 shows the demographic profile of the students. The gender distribution of the students was quite balanced, and all respondents were between the ages of 21 to 27.

**Table 1.** Demographic profile of respondents (N = 301).

| Demographic Variables | | Number | Percentage |
|---|---|---|---|
| Gender | Male | 131 | 43.52 |
| | Female | 170 | 56.48 |
| Education | Undergraduate | 245 | 81.40 |
| | Postgraduate | 56 | 18.60 |
| Field of study | Linguistics | 96 | 31.90 |
| | Psychology | 64 | 21.26 |
| | Law | 141 | 46.84 |

The online survey aimed to define the factors inducing greater intention for students to choose a blended-learning format was conducted on Google. The Likert-type five-point scale was used to measure the variables. The basis for the items measuring "perceived usefulness", "satisfaction", and "confirmation" were the scales from Bhattacherjee [12] and Ifinedo [55]. To evaluate attitude, subjective norms, and perceived behavioral control, we used four items, each adopted from Fishbeinand and Ajzen [56], Lung-Guang [57], and Yeap et al. [58]. To measure the continuance intention to learn, three items were adopted from Lung-Guang [57] and Yeap et al. [58]. The items used in the online survey are presented in Appendix A.

To empirically test the proposed model (Figure 1) and assess the proposed hypothesis, a quantitative research method was performed. In order to conduct the analysis, PSS 24.0 and SmartPLS 3.0 programs were used. Accordingly, a cross-sectional data collection approach using an online questionnaire was used to empirically test the model and identify structural relationships between reflexive latent constructs.

## 3. Results

We started with checking the normality of the data using the Shapiro–Wilk test, which showed a significant result. In addition, the adequacy of sample was validated using Kaiser–Meyer–Olkin (KMO) tests and Bartlett's sphericity test, which showed positive results (KMO: 0.889; Bartlett's test: significance at 0.00). Pearson correlation analysis is presented in Table 2.

**Table 2.** Correlation analysis.

| Indicator | CON | SAT | PU | ATD | SN | PBC | CIL |
|---|---|---|---|---|---|---|---|
| CON | 1 | | | | | | |
| SAT | 0.538 ** | 1 | | | | | |
| PU | 0.432 ** | 0.532 ** | 1 | | | | |
| ATD | 0.370 * | 0.675 ** | 0.376 * | 1 | | | |
| SN | 0.209 | 0.401 ** | 0.602 ** | 0.317 * | 1 | | |
| PBC | 0.485 ** | 0.326 * | 0.531 ** | 0.451 ** | 0.610 ** | 1 | |
| CIL | 0.375 * | 0.403 ** | 0.414 ** | 0.405 ** | 0.437 ** | 0.515 ** | 1 |

Note: * $p < 0.05$; ** $p < 0.01$; CON-confirmation, SAT-satisfaction, PU-perceived usefulness, ATD-attitude, SN-subjective norms, PBC-perceived behavioral control, CIL-continuance intention to learn.

According to the results, a positive correlation was detected between all studied indicators. The strongest relationship was revealed between attitude and satisfaction, while such indicators as subjective norms and confirmation correlate weakly (R = 0.209).

For the research model testing, we used Partial Least Squares (PLS), as this technique has a number of advantages. For instance, SEM (Structural Equation Modeling) as a second-generation method can analyze indicator loads (and weights) on structures (therefore measuring the reliability of a structure) and evaluate random relationships with structures in multistage models [59]. Also, PLS is a robust covariance modeling for theory testing that is appropriate for our study. Further, this technique provides a better approximation with regard to the final estimates [60]. Thus, PLS was chosen to test the exploratory model of

this study. The data was analyzed in two steps (measurement model and structural model). The measurement model was confirmed by the establishment of reliability and validity. Therefore, the research hypotheses were tested using the structural model.

First, we performed the reliability analysis using Cronbach's $\alpha$ and Research Unit Reliability (C.R.) to measure the internal consistency of the variables used in the research. According to Table 3, the internal consistency of the variables used is normal ($0.879 < \alpha < 0.927$; $0.889 < $ C.R. $ < 0.949$). The reliability between the measurement items was more than 0.70 (standard value).

**Table 3.** Measurement model.

| Indicator | Items | Factor Loadings | $\alpha$ | C.R. | AVE |
|---|---|---|---|---|---|
| CON | CON1 | 0.811 | 0.879 | 0.901 | 0.807 |
| | CON2 | 0.819 | | | |
| | CON3 | 0.832 | | | |
| | CON4 | 0.819 | | | |
| SAT | SAT1 | 0.817 | 0.882 | 0.889 | 0.790 |
| | SAT2 | 0.808 | | | |
| | SAT3 | 0.795 | | | |
| PU | PU1 | 0.804 | 0.895 | 0.910 | 0.816 |
| | PU2 | 0.821 | | | |
| | PU3 | 0.817 | | | |
| ATD | ATD1 | 0.889 | 0.903 | 0.922 | 0.889 |
| | ATD2 | 0.905 | | | |
| | ATD3 | 0.873 | | | |
| SN | SN1 | 0.845 | 0.897 | 0.907 | 0.756 |
| | SN2 | 0.834 | | | |
| | SN3 | 0.821 | | | |
| PBC | PBC1 | 0.906 | 0.927 | 0.949 | 0.889 |
| | PBC2 | 0.904 | | | |
| | PBC3 | 0.911 | | | |
| | PBC4 | 0.917 | | | |
| CIL | CIL1 | 0.807 | 0.901 | 0.914 | 0.804 |
| | CIL2 | 0.826 | | | |

To evaluate the instrument reliability and validity, we applied the confirmatory factor analysis (CFA) approach for our seven-factor research model. Tables 3 and 4 present results of the analysis.

**Table 4.** Discriminant validity.

| Indicator | CON | SAT | PU | ATD | SN | PBC | CIL |
|---|---|---|---|---|---|---|---|
| CON | 0.855 | | | | | | |
| SAT | 0.383 * | 0.903 | | | | | |
| PU | 0.87 * | 0.319 * | 0.817 | | | | |
| ATD | 0.400 ** | 0.415 ** | 0.462 ** | 0.901 | | | |
| SN | 0.527 ** | 0.375 ** | 0.218 * | 0.419 ** | 0.863 | | |
| PBC | 0.161 * | 0.152 * | 0.303 * | 0.391 * | 0.365 * | 0.879 | |
| CIL | 0.348 * | 0.167 * | 0.287 * | 0.294 * | 0.211 * | 0.542 ** | 0.877 |

Note: * $p < 0.05$; ** $p < 0.01$.

Convergence validity is indicated as a high correlation between the same concepts. The factor loadings were found between 0.795 and 0.917, and the AVE value was 0.756 or more, confirming the convergence validity of research units in Table 3. In addition, among the latent variables, the square root of AVE in each construct was greater than the

other correlation values in Table 2. Thus, it can be concluded that discriminant validity is well established.

In addition, due to the cross-loading criterion, there is the presence of discriminant validity between all constructs, as the loading indicators on its own construct are in all cases higher than all its cross-loadings with other constructs (Table 5).

**Table 5.** Cross-loading criterion.

| Constructs | CON | SAT | PU | ATD | SN | PBC | CIL |
|---|---|---|---|---|---|---|---|
| CON1 | **0.867** | 0.402 | 0.396 | 0.385 | 0.117 | 0.321 | 0.414 |
| CON2 | **0.879** | 0.462 | 0.403 | 0.452 | 0.181 | 0.216 | 0.439 |
| CON3 | **0.891** | 0.413 | 0.456 | 0.462 | −0.017 | 0.198 | 0.461 |
| CON4 | **0.904** | 0.501 | 0.388 | 0.418 | 0.033 | 0.201 | 0.396 |
| SAT1 | 0.542 | **0.911** | 0.514 | 0.512 | 0.092 | 0.086 | 0.512 |
| SAT2 | 0.479 | **0.874** | 0.453 | 0.564 | 0.147 | 0.168 | 0.476 |
| SAT3 | 0.502 | **0.893** | 0.417 | 0.495 | 0.207 | 0.092 | 0.416 |
| PU1 | 0.453 | 0.514 | **0.865** | 0.478 | 0.167 | 0.179 | 0.389 |
| PU2 | 0.511 | 0.396 | **0.879** | 0.439 | 0.109 | 0.093 | 0.401 |
| PU3 | 0.413 | 0.453 | **0.894** | 0.501 | 0.192 | 0.116 | 0.373 |
| ATD1 | 0.417 | 0.312 | 0.517 | **0.911** | 0.107 | 0.189 | 0.415 |
| ATD2 | 0.387 | 0.379 | 0.477 | **0.901** | 0.052 | 0.204 | 0.433 |
| ATD3 | 0.471 | 0.401 | 0.418 | **0.879** | 0.04 | 0.116 | 0.385 |
| SN1 | −0.018 | 0.167 | 0.110 | 0.119 | **0.792** | 0.341 | 0.226 |
| SN2 | −0.031 | 0.017 | 0.085 | 0.205 | **0.844** | 0.358 | 0.215 |
| SN3 | −0.019 | 0.092 | 0.032 | 0.116 | **0.887** | 0.311 | 0.183 |
| PBC1 | 0.021 | 0.178 | 0.201 | 0.092 | 0.118 | **0.816** | 0.179 |
| PBC2 | 0.127 | 0.211 | 0.119 | 0.153 | 0.074 | **0.895** | 0.201 |
| PBC3 | 0.173 | 0.169 | 0.176 | 0.148 | 0.039 | **0.879** | 0.239 |
| PBC4 | 0.092 | 0.197 | 0.206 | 0.11 | 0.138 | **0.814** | 0.268 |
| CIL1 | 0.435 | 0.459 | 0.398 | 0.416 | 0.236 | 0.316 | **0.891** |
| CIL2 | 0.389 | 0.506 | 0.427 | 0.501 | 0.265 | 0.362 | **0.904** |

Also, the proportion of the variance explained ($R^2$), as suggested by Hair et al. [60], was calculated to evaluate the predictive power criterion of a structured model to investigate its quality (Table 6).

**Table 6.** The predictive power of each factor in the measured model.

| Factors | CON | SAT | PU | ATD | SN | PBC | CIL |
|---|---|---|---|---|---|---|---|
| $R^2$ | 0.76 | 0.80 | 0.78 | 0.90 | 0.62 | 0.70 | 0.82 |
| Predictive power | substantial | substantial | substantial | substantial | moderate | moderate | substantial |

The predictive power is described as substantial, moderate, and weak, with $R^2 > 0.75$ or 0.50 or 0.25. The results show that subjective norms and perceived behavioral control have moderate predictive power, while the factors including confirmation, satisfaction, perceived usefulness, and attitude have substantial predictive power in the study.

The SmartPLS calculated coefficients are shown in Table 7. As Table 7 displays, confirmation has a significant effect on satisfaction, positively influencing PU and CIL, while it is significant towards SAT and PU, which in turn have a positive impact on continuance intention to learn in blended environment. Thus, the hypotheses H1a, H1b, H2, and H3a were supported.

**Table 7.** Standardized structural estimates.

| Hypothesis | Path | Mean | SD | β | t-Value | p-Value | Results |
|---|---|---|---|---|---|---|---|
| H1a | CON→SAT | 0.594 | 0.047 | 0.661 | 15.411 | 0.001 | Supported |
| H1b | CON→PU | 0.352 | 0.037 | 0.361 | 9.877 | 0.000 | Supported |
| H2 | SAT→CIL | 0.224 | 0.031 | 0.230 | 7.790 | 0.001 | Supported |
| H3a | PU→CIL | 0.474 | 0.041 | 0.457 | 8.189 | 0.000 | Supported |
| H3b | PU→ATD | 0.416 | 0.038 | 0.417 | 7.819 | 0.000 | Supported |
| H4 | ATD→CIL | 0.587 | 0.046 | 0.691 | 15.917 | 0.000 | Supported |
| H5 | SN→CIL | 0.153 | 0.052 | 0.152 | 2.314 | 0.032 | Supported |
| H6 | PBC→CIL | 0.107 | 0.057 | 0.107 | 2.021 | 0.006 | Supported |

Note: *p*-value < 0.000—significant at 1% level; *p*-value < 0.05—significant at 5% level.

As shown in Table 5, PU has a positive influence on ATD (β = 0.417, t = 7.819, $p$ < 0.01); thus, H3b was supported. Hypothesis H4, H5, and H6 were supported because ATD (β = 0.391, t = 9.811, $p$ < 0.05), SN (β = 0.152, t = 2.314, $p$ < 0.05) and PBC (β = 0.107, t = 2.021, $p$ < 0.05) have significant effects on CIL. The explanatory power (R2) of CIL is 0.711.

## 4. Discussion

This study has its own characteristics that distinguish it from many earlier works. The main difference is our own approach to research. This study was conducted on the basis of two models—ECM [12] and TPB [13]. Previously developed models by such researchers as Ajzen [13] and Bhattacherjee [12] formed the basis of many studies [32–34,44,45,47]. In this article, these models were combined and considered comprehensively. The intent to learn in a blended environment during a pandemic depends on the influence of each indicator presented in both models. Many studies [43,50,51] use the proposed models in the field of trade to analyze the behavior of buyers; however, both of these models are less often used in the field of education. It seems important to us to explore the intention towards blended learning, as blended learning has become firmly entrenched in education against the backdrop of the pandemic.

The main goal of this study was to determine the factors that have an influence on the continuance intention to learn in blended environments. Theoretically based on ECM and TPB, this study tested eight hypotheses related between the key components of these theories. According to the results of this study, all the proposed hypotheses were confirmed, confirming the influencing power of research model indicators. Also, it was revealed that such indicators as confirmation and attitude are the main factors that have an effect on the intention to learn in blended environment.

According to the results of correlation analysis, a positive correlation was detected between all studied indicators. The strongest relationship was revealed between attitude and satisfaction, while such indicators as subjective norms and confirmation correlate weakly. The results of this research show that subjective norms and perceived behavioral control have moderate predictive power, while the factors including confirmation, satisfaction, perceived usefulness, and attitude have substantial predictive power in the study. Thus, we conclude that confirmation significantly and positively affects satisfaction and perceived usefulness. Perceived usefulness significantly and positively affects attitude. Attitude, satisfaction, and perceived usefulness in turn significantly and positively affect continuance intention to learn in blended environment. Subjective norms and perceived behavioral control have moderate influence, but also significant. Therefore, when introducing a blended learning environment, faculty should keep in mind that the blended environment must be properly organized. It is important because first impressions and experiences of students learning in new environments will determine their confirmation, satisfaction, attitude, and other factors, influencing continuous intention to learn. Consequently, blended environments should be comfortable for use, should contain all important information, and clear instructions. The quality of the technology is a strong contributor to learner attitude and satisfaction in online learning.



The conclusions of the study have practical implications and can be used by man-agers and faculty of the universities who have the ability to increase students' positive attitudes towards blended learning environment, as attitude directly influences the continuance intention to learn.

Our study has limitations. The study was conducted only in Russia, while the COVID-19 pandemic and blended learning are common worldwide. In addition, the study included only humanities students. Education in technical areas has its own characteristics, and the intention for blended learning for technical specialties may differ.

In our further research, we are going to implement a comparative analysis on differences between the students' continuance intention to learn in a blended environment (gender, level of education, field of study).

**Author Contributions:** Conceptualization, A.K. and T.B.; methodology, E.T. and A.K.; software, T.B.; validation, E.T.; formal analysis, A.K. and E.T.; investigation, A.K. and E.T.; resources, T.B.; data curation, T.B.; writing—original draft preparation, A.K. and E.T.; writing—review and editing, A.K.; visualization, E.T.; supervision, A.K.; project administration, T.B. and T.B.; funding acquisition, A.K. All authors have read and agreed to the published version of the manuscript.

**Funding:** The research is partially funded by the Ministry of Science and Higher Education of the Russian Federation under the strategic academic leadership program 'Priority 2030' (Agreement 075-15-2021-1333 dated 30 September 2021).

**Informed Consent Statement:** Informed consent was obtained from all subjects involved in the study.

**Conflicts of Interest:** The authors declare no conflict of interest.

## Appendix A

Students measured "To what extent do you agree or disagree with the following statements".

**Table A1.** Students' survey.

| № | Construct | Scale |
|---|---|---|
| 1 | Confirmation | CON 1 My experience with using the blended learning system was better than I expected<br>CON 2 The service level provided by the blended learning system was better than I expected<br>CON 3 The blended learning systems can meet demands in excess of what I required for the service<br>CON 4 The effectiveness of the blended learning course was better than I expected |
| 2 | Satisfaction | SAT 1 My overall experience with blended learning was very satisfied<br>SAT 2 My overall experience with blended learning was very pleased<br>SAT 3 My overall experience with blended learning was very contended |
| 3 | Perceived usefulness | PU 1 I believe that using blended learning technologies would improve my ability to learn<br>PU 2 I believe that blended learning technologies would allow me to get my work done more quickly<br>PU 3 I believe that blended format would be useful for my learning |
| 4 | Attitude | ATD 1 I would like my coursework more if I used blended learning<br>ATD 2 Using blended learning in my coursework would be a pleasant experience<br>ATD 3 Using blended learning in my coursework would be a wise idea |

**Table A1.** *Cont.*

| № | Construct | Scale |
|---|-----------|-------|
| 5 | Subjective norms | SN 1 Most people who are important to me think that it would be fine to use a blended learning technology for university courses<br>SN 2 I think other students in my classes would be willing to adopt a blended learning technology<br>SN 3 Most people who are important to me would approve of using a blended learning technology for university courses |
| 6 | Perceived behavioral control | PBC 1 I have sufficient extent of knowledge to use blended learning<br>PBC 2 I have sufficient extent of control to make a decision to adopt blended learning<br>PBC 3 I have sufficient extent of self-confidence to make a decision to adopt blended learning<br>PBC 4 I would be able to use the blended learning system well for learning process |
| 7 | Continuance intention to learn in blended environment | CIL 1 I will strongly recommend that others use blended learning<br>CIL 2 I intend to learn in blended format in the future |

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
