# Peer review of "Factors Influencing Students’ Continuance Intention to Learn in Blended Environments at University"

_electronics, doi:10.3390/electronics11132069_

Round 1

Reviewer 1 Report

The paper is interesting and quite well-written.

There are some points that need to be addressed

In the section Theoretical Background

The hypotheses are not below the relevant theoretical section but they are all at the end of the section. You have to argue about the use of the constructs and in the end, present the hypothesis.

In the results section:

Table 3. For each of the items in the construct give the Weights or loadings. (Not the range)

Discriminant validity–loading and cross loading criterion was not checked. You can see:

Rezaei, S., Ali, F., Amin, M. and Jayashree, S., 2016. Online impulse buying of tourism products: The role of web site personality, utilitarian and hedonic web browsing. Journal of Hospitality and Tourism Technology.

The validated model was not given. Also the quality of the model under investigation is not so good. See for example the figures in the paper:

Ali, F., Rezaei, S., Hussain, K. and Ragavan, N.A., 2014. International business travellers’ experience with luxury hotel restaurants: the impact of foodservice experience and customer satisfaction on dining frequency and expenditure. International Journal of Hospitality and Event Management, 1(2), pp.164-186.

In the conclusion section

Conclusions are very poor. You have to explain and discuss the effect of each variable on the influence on the continuance intention to learn in blended environment. You only give the findings, but this is not enough. Implications are even poorer. You mention “The conclusions of the study have practical implications and can be used by managers and faculty of the universities who have abilities to increase students' positive attitudes towards blended learning environment as attitude influences directly the continuance intention to learn” but nothing really about the practical implications. Without conclusions and implications, I am really wondering about the worth of the paper.

Reviewer 2 Report

The authors are concerned with blended learning. They study the factors that influence the intention to continue learning in a blended environment.

Baranova et al. propose a research model based on the expectation-confirmation model and the theory of planned behavior. A case study has allowed the authors to analyze and test the stated hypotheses. In this sense, the influencing power of research model indicators were confirmed.

The motivation and justification of the paper are appropriate.

Finally, I include some typographical errors and recommendations:

In line 271,

For: There-fore,

read: Therefore,

In line 303,

For: (β==0.391

read: (β=0.391

Complete ref. 3:

University Responses to Covid-19

Author: Louise Bena

Complete ref. 10:

https://doi.org/10.1016/j.ijedro.2020.100012

Complete ref. 11:

https://doi.org/10.1016/j.childyouth.2020.105578

Complete ref. 12:

https://doi.org/10.2307/3250921

Complete ref. 13:

https://doi.org/10.1016/0749-5978(91)90020-T

In ref. 14:

For: Iap, 2009

read: IAP, Information Age Publishing, 2009

Complete ref. 19:

https://doi.org/10.1080/02188791.2020.1766417

In ref. 26:

For: International journal of educational technology in Higher education

read: International Journal of Educational Technology in Higher Education

In ref. 41:

For: Asian transport studies

read: Asian Transport Studies

In ref. 44:

For: planned behavior 1

read: planned behavior

https://doi.org/10.1111/j.1559-1816.2002.tb00236.x

In ref. 49:

For: Organizational behavior and human decision processes

read: Organizational Behavior and Human Decision Processes

In ref. 49:

https://doi.org/10.1287/isre.1050.0042

In ref. 55:

Students’perceived

Students’ perceived

In ref. 57:

https://doi.org/10.1016/j.compedu.2019.02.004

Reviewer 3 Report

The work is intriguing and up-to-date. The paper is structured and written in an academic style. However, it needs minor English language improvements and corrections of typos (e.g., l. 102, 271, 278). Further, resolving the following issues could improve its quality and strengthen the research impact.

  1. The main concern is that the research topic seems to be scientifically related to the educational area and blended learning in particular. However, there is no relation to the scope of the journal Electronics. The paper does not make reference to some of the Electronics journal topics. I would suggest transferring it to a journal with a scope more relevant to the presented topic.
  2. The authors should clearly outline the contribution of the research in the introduction section.
  3. The authors should add a paragraph with related works concerning previous studies on the topic and further discuss and compare them with the findings of this study.
  4. There is a discrepancy in the description of the research model – in the Abstract: “… involves a total of seven latent constructs and contains a total of eleven hypotheses” (l. 13-14), while in section 1.1.5 is written, “… involves a total of seven latent constructs and consists of eight hypotheses” (l. 226-227). This issue should be resolved.
  5. This paper involves an online survey consisting of 22 items (l. 16), which is said to be based on several surveys, from which a different number of items are adopted (l. 242-246). The whole survey is not presented, so it should be included as an Appendix to give the readers a more comprehensive understanding.
  6. The authors should explain better the materials and methods used before presenting the results.
  7. The results could be commented on more meaningfully.
  8. The discussion part should include comparisons with surveys on similar topics.
  9. Extended conclusions could benefit the research.

Round 2

Reviewer 1 Report

I think that the authors have responded adequately to my comments.

Well done

Reviewer 2 Report

The authors are concerned with blended learning. They study the factors that influence the intention to continue learning in a blended environment. Baranova et al. propose a research model based on the expectation-confirmation model and the theory of planned behavior. A case study has allowed the authors to analyze and test the stated hypotheses. In this sense, the influencing power of research model indicators were confirmed.

The motivation and justification of the paper are appropriate.

In my humble opinion, the authors have improved the manuscript. The authors have satisfactorily resolved the reviewers' indications.